# The Crucial Roles of Bmi-1 in Cancer: Implications in Pathogenesis, Metastasis, Drug Resistance, and Targeted Therapies

**DOI:** 10.3390/ijms23158231

**Published:** 2022-07-26

**Authors:** Jie Xu, Lin Li, Pengfei Shi, Hongjuan Cui, Liqun Yang

**Affiliations:** 1State Key Laboratory of Silkworm Genome Biology, Southwest University, Chongqing 400716, China; xujiesw@sina.com (J.X.); lilin1591905507@126.com (L.L.); spf2019@sina.cn (P.S.); 2Cancer Center, Medical Research Institute, Southwest University, Chongqing 400716, China

**Keywords:** Bmi-1, miRNAs, tumorigenesis, chemoresistance, cancer therapy, molecular mechanism

## Abstract

B-cell-specific Moloney murine leukemia virus integration region 1 (Bmi-1, also known as RNF51 or PCGF4) is one of the important members of the PcG gene family, and is involved in regulating cell proliferation, differentiation and senescence, and maintaining the self-renewal of stem cells. Many studies in recent years have emphasized the role of Bmi-1 in the occurrence and development of tumors. In fact, Bmi-1 has multiple functions in cancer biology and is closely related to many classical molecules, including Akt, c-MYC, Pten, etc. This review summarizes the regulatory mechanisms of Bmi-1 in multiple pathways, and the interaction of Bmi-1 with noncoding RNAs. In particular, we focus on the pathological processes of Bmi-1 in cancer, and explore the clinical relevance of Bmi-1 in cancer biomarkers and prognosis, as well as its implications for chemoresistance and radioresistance. In conclusion, we summarize the role of Bmi-1 in tumor progression, reveal the pathophysiological process and molecular mechanism of Bmi-1 in tumors, and provide useful information for tumor diagnosis, treatment, and prognosis.

## 1. Introduction

Polycomb group (PcG) proteins are a group of transcriptional inhibitors that regulate targeted genes at the chromatin level. They also play a critical role in embryonic development, cell proliferation, and tumorigenesis [1,2]. PcG proteins act on the development of organisms by forming two multimeric protein complexes: the polycomb repressive complex 1 (PRC1) and the polycomb repressive complex 2 (PRC2)) [3]. B-cell-specific Moloney murine leukemia virus integration region 1 (Bmi-1, also known as RNF51 or PCGF4) is one of the core members of the PRC1 complex. This complex, acting through chromatin remodeling, is an essential epigenetic repressor involved in embryonic development and self-renewal in somatic stem cells [4,5,6]. Therefore, Bmi-1 also is an oncogene, and its abnormal expression is associated with tumorigenesis and drug resistance in many cancers, including bladder cancer cells, B-cell lymphoma cells, melanoma cells, and others. Noncoding RNAs (ncRNAs) are a class of RNAs produced in noncoding regions that are not translated into proteins but are widely expressed in organisms [7]. With the development of gene sequencing technology, increasing evidence suggests that ncRNAs have crucial biological functions and play core roles in regulating gene expression [8]. In recent years, it has been identified that microRNAs (miRNAs) play important roles in various biological processes by inhibiting the translation of mRNAs and inducing their degradation, which makes them potential molecular targets for cancer therapy [9,10,11,12,13]. Interestingly, several recent studies have found that Bmi-1 is targeted and regulated by many miRNAs in cancer. Furthermore, the unique biological functions and cancer-promoting effects of Bmi-1 have also attracted increased attention. Therefore, in this review, we first describe the normal function of Bmi-1; then, we review the relevant signaling pathways regulated by Bmi-1 in normal and carcinogenic conditions and its regulatory network with miRNAs, to provide a reference for a more comprehensive understanding of the role of Bmi-1 in cancer.

## 2. Molecular Features and Characteristics of Bmi-1

The structure of Bmi-1 in humans and mice is very similar, and the homologies of DNA and amino acid in Bmi-1 are 86% and 98%, respectively. The *Bmi-1* gene in humans, composed of 10 exons and 9 introns, is localized on the short arm of chromosome 10 (10p11.23). The Bmi-1 protein is 37 kD in size, consists of 326 amino acids, and has a highly conserved structure. There are several important structures in the amino acid sequence of Bmi-1 protein. The N-terminal is a RING finger domain, which is composed of a zinc finger and C3hC4 sequence. Bmi-1 can regulate transcription, affect cell proliferation, and participate in the formation of the malignant tumors by RING finger binding to other critical proteins. The helix–turn–helix structure (HTH structure) is located in the center of the Bmi-1 protein, and mediates the transcriptional inhibition of Bmi-1 by binding with DNA. At the carboxyl terminus, the PEST structure of the PEST region situated at the C terminal contains many serine, glutamic acid, threonine, and proline residues [14], which are associated with intracellular turnover of Bmi-1 protein. Two nuclear localization signals (NLS1, NLS2) are contained in Bmi-1, of which NLS2 is required for Bmi-1 localization in the nucleus. Figure 1 illustrates the structure of the *Bmi-1* gene and Bmi-1 protein, the illustration is redrawn (with modifications) from Sahasrabuddhe (2016) [15].

## 3. Upstream Regulatory Mechanisms of Bmi-1

Bmi-1 is expressed in almost every human tissue and in many cancers, and serves as a biomarker for some cancers. Therefore, it is crucial to explore the mechanisms that regulate *Bmi-1* mRNA and Bmi-1 protein levels. Over the past decade, researchers have made some progress in the regulation of Bmi-1 at the transcriptional, post-transcriptional, and post-translational levels. It is currently known that *Bmi-1* is included in the positive or negative transcriptional regulation of many transcription factors. Positively regulated transcription factors are *Sp1*, *Twist1*, *FoxM1*, *ZEB1*, *E2F1*, *SALL4*, *MYC-N*, *c-MYC*, and *HDACs*, while negatively regulated transcription factors are *Mel18*, *Nanog*, and *KLF4* [15].

Post-translational modifications (PTMs) of proteins are an important part of proteomics. Proteins change their spatial conformation, activity, stability, and interaction performance with other molecules through PTMs, thereby participating in the regulation of various life activities in the body. Most proteins can undergo PTMs, and there are more than 200 known covalent modifications of proteins, including phosphorylation, nitrosylation, nitration, ubiquitination, and sumoylation, among others. PTMs are associated with many important diseases, including cancer. There have been some studies on the post-translational modification of Bmi-1.Bmi-1 can be activated by phosphorylation. For example, the state of Bmi-1 protein binding to chromosomes is related to its phosphorylation; in G1/S phase, non-phosphorylated Bmi-1 specifically binds to chromosomes, while in G2/M phase, phosphorylated Bmi-1 does not bind to chromosomes [16]. Bmi-1 represses the expression of the Ink4a-Arf locus, which encodes two tumor suppressor genes, p16^Ink4a^ and p19^Arf^. Bmi-1 is phosphorylated (and inactivated) at Ser 316 by Akt serine/threonine kinase, also known as protein kinase B (PKB), to promote p16^Ink4a^ and p19^Arf^ expression and inhibit hematopoietic stem cell self-renewal [17]. Similarly, the Wnt5a-ROR1 complex can phosphorylate Bmi-1 through the Akt pathway to promote tumorigenesis [18]. It has also been reported that activation or overexpression of MAPKAP kinase 3 leads to the phosphorylation of Bmi-1 and other PcG members, and promotes their dissociation from chromatin [19]. Sumoylation is a reversible post-translational modification which has emerged as a crucial molecular regulatory mechanism, involved in the regulation of replication, cell-cycle progression, protein transport and the DNA damage response. The study found that DNA damage can induce the sumoylation of lysine 88 in Bmi-1, and knockout of chromobox 4 (CBX4), a member of the polycomb group family of epigenetic regulatory factors, can eliminate this phenomenon, which means that CBX4 can mediate the sumoylation of Bmi-1 [20]. Moreover, p53/p21 can inversely regulate Bmi-1 expression by affecting ubiquitin/proteasome activity [21]. This also implies that Bmi-1 may be ubiquitinated. Furthermore, Ser255 was found to be the site of O-GlcNAcylation of Bmi-1 in prostate cancer, and O-GlcNAcylation of Bmi-1 promoted the stability of Bmi-1 protein and its oncogenic activity [22].

MicroRNAs (miRNAs or miRs) are a class of small of ncRNAs that contain about 19–24 nt. Although miRNAs do not encode proteins, they regulate the expression levels of some genes at the post-transcriptional stage. MiRNAs negatively regulate the expression of target genes by complete or incomplete complementary binding to the 3′-UTR of target mRNAs, promoting the degradation or translational repression of target mRNAs. There is growing evidence that miRNAs are involved in essential biological processes, including development, differentiation, proliferation, apoptosis, invasion, and metastasis. The 3′UTR length of Bmi-1 is 2090 nt. The miRNAs mainly target two spacers of the 3′UTR of Bmi-1. The first target position is 469–725, which includes miR-128, miR-221, miR-183, and miR-200b/c; the other position is 1442–1758, including miR-203, miR-218. We summarize the interaction between Bmi-1 and miRNAs in human tumors in Table 1, to provide information for further research on cancer treatment and clinical applications.

## 4. Bmi-1-Targeted Processes in Cancer

Bmi-1 protein belongs to the polycomb family of transcriptional repressors, and binds to polycomb response elements in the genome to silence the transcription of specific target genes. The genes to be transcriptionally regulated by Bmi-1 include *INK4a*/*ARF*, *Pten*, homeobox protein HoxC13, human telomerase reverse transcriptase (*hTERT*), etc. [76,77,78,79]. Figure 2 illustrates the downstream genes directly regulated by Bmi-1. In addition, Bmi-1 is involved in many classical signaling pathways, for example mTOR, NF-κB, PI3K/Akt, and other signaling pathways. The function of Bmi-1 in tumors has been identified in various pathological processes, including abnormal cell proliferation, evasion of apoptosis, migration of cancer cells, and stemness maintenance of cancer stem cells (CSCs). Therefore, we provide a corresponding review of the key target genes and signaling pathways behind these processes.

### 4.1. Bmi-1 in Cancer Proliferation

Hyperproliferation is a typical feature of cancer progression, manifested by altered expression and activity of cell cycle-related proteins. The most classic downstream target of Bmi-1 is INK4a/ARF. The tumor suppressor gene INK4a/ARF can encode two regulatory genes: *p16^INK4a^* and *p14^ARF^* (*p19^ARF^* in mice). Their normal and orderly expression is the key to maintaining the balance of the cell cycle. On the one hand, P16^INK4a^ arrests cells in the G_0_/G_1_ phase through cyclinD-CDK4/6-pRb-E2F. Mechanistically, when Bmi-1 is deficient, p16^Ink4a^ is up-regulated, which prevents the binding of CDK4/6 to cyclinD, leading to phosphorylation of Rb. Phosphorylated Rb binds to E2F transcription factor to inhibit E2F transcription-factor-mediated transcription, which arrests the cell cycle in the G0/G1 phase [76]. On the other hand, p14^ARF^ regulates the cell cycle through the MDM2/p53 pathway [15,84,85,86,87]. In detail, the p14^ARF^ (p19^ARF^) protein can antagonize the ubiquitin-protein ligase MDM2 and p53-dependent transcription, thereby stabilizing tumor protein p53, causing cell apoptosis [15,84,85,86,87]. Bmi-1-shRNA reduces the expression of cyclin D1 and increases the expression of cyclin dependent kinase inhibitor p21 and p27 through the p16^Ink4a^ independent pathway, to arrest lung adenocarcinoma cells in the G0/G1 phase [88]. *USP22*, an oncogene, could actively and effectively participate in the regulation of the cell cycle through the INK4a/ARF signaling pathway mediated by Bmi-1 in human colorectal cancer cells [89]. Similar phenomena have also been detected in gallbladder carcinoma, HeLa cells, lung cancer, esophageal carcinoma cell, etc. [90,91,92,93]. Bmi-1 can also activate the PI3K/mTOR/4EBP1 signaling pathway in ovarian cancer cells to regulate cell proliferation [94]. N-acetylglucosamine transferase (OGT) is the only known enzyme to catalyze the O-GlcNAcylation in humans. O-GlcNAcylation can promote the stability and oncogenic activity of Bmi-1 [22]. MiR-485-5p can regulate the O-GlcNAcylation level and the stability of Bmi-1 by inhibiting OGT, and then inhibit the proliferation of colorectal cancer cells [28]. Nevertheless, some studies have shown that Bmi-1 does not affect the cell cycle of lung cancer cells and the expression of p16/p19, PTEN, Akt, or p-Akt [95]. This may be related to cell type.

### 4.2. Bmi-1 in Cancer Apoptosis

Strong anti-apoptotic ability is one of the characteristics of cancer cells. Bmi-1 directly and indirectly regulates cell apoptosis via various pathways. The first pathway is the p14^ARF^/MDM2/p53 signal pathway mentioned above. Inhibition of Bmi-1 can promote the expression of *p14^ARF^* (*p19^ARF^*), and p14^ARF^ (p19^ARF^) can antagonize the ubiquitin-protein ligase MDM2 to stabilize p53 and cause apoptosis. This phenomenon has been reported in many studies [29,96,97]. A second pathway involves the inhibition of Bmi-1, causing abnormal mitochondrial function and increasing the level of ROS, leading to apoptosis. For example, a recent study found that sodium butyrate induced ROS-mediated apoptosis by inhibiting the expression of Bmi-1 through miR-139-5p [13]. The third pathway is the Bmi-1 regulation of cell apoptosis by the NF-κB pathway. Expression of Bmi-1 could protect glioma cells from apoptosis by activating the NF-κB pathway [98]. Consistently, overexpression of Bmi-1 can increase cisplatin-induced apoptosis resistance in osteosarcoma by activating the NF-κB signal pathway [99]. In addition, the loss of Bmi-1 leads to impaired repair of DNA double-strand breaks by homologous recombination, ultimately leading to apoptosis [96]. Inhibition of Bmi-1 reduces the ubiquitination of myeloid cell leukemia sequence 1 (Mcl-1) by downregulating deubiquitinating enzyme (DUB3), resulting in cancer cell apoptosis [99].

### 4.3. Bmi-1 in Cancer Autophagy

Autophagy is a conserved self-degradation system that is critical for maintaining cellular homeostasis during stress conditions. Autophagy plays a dichotomous role in cancer by suppressing benign tumor growth but promoting advanced cancer growth [100]. Bmi-1 also affects cellular autophagy generally through PI3K/Akt and AMPK signaling pathways. Knockdown of Bmi-1 induced autophagy in ovarian cancer cells via ATP depletion [101]. Sodium butyrate induced AMPK/mTOR pathway-dependent autophagy via the miR-139-5p/Bmi-1 axis in human bladder cancer cells [13]. Knockdown of Bmi-1 in breast cancer cells also induced autophagy [102,103]. Additionally, betulinic acid induced autophagy and apoptosis in bladder cancer cells through the Bmi-1/ROS/AMPK/mTOR/ULK1 axis [86]. Similarly, Bmi-1 overexpression promoted migration and proliferation of cardiac fibroblasts by regulating the Pten/PI3K/Akt/mTOR signaling pathway [104]. In addition, the expression of Bmi-1 targets the inhibition of cyclinG2 expression, and cyclinG2 acts by disrupting the phosphatase 2A complex, which activates the PKCζ/AMPK/JNK/ERK pathway involved in autophagy [105].

### 4.4. Bmi-1 in Cancer EMT

Cancer cells have the characteristics of invasion into surrounding tissues and metastasis to distant tissues, and they continue to grow in these organs, destroying the function of the organs, and finally causing the death of the patient. The expression of Bmi-1 is closely related to tumor migration and invasion [93,95,106,107,108,109]. It has been reported that down-regulation of Bmi-1 reduces the migration and invasion of endometrial cancer cells in vivo and in vitro, by increasing the expression levels of E-cadherin and keratin, and down-regulating N-Cadherin, vimentin, and SLUG [106]. Likewise, Bmi-1 induces miR-27a and miR-155 to negatively regulate the expression of raf kinase inhibitory protein (RKIP), which promotes the migration and invasion of gastric cancer cells [107]. Additionally, Bmi-1 promotes the migration and invasion of Hepatocellular carcinoma by inhibiting phosphatase and tensin homolog (PTEN) and activating the PI3K/Akt pathway, and increasing the expression of MMP2, MMP9 and VEGF [110]. Similar phenomena have also been found in breast cancer, colon cancer, esophageal cancer cells [107]. Moreover, Bmi-1 increases luciferase activity of the MMP9 promoter-driven reporter gene containing an NF-κB binding site, which promotes MMP9 transcription levels in glioma cells [111]. Bmi-1 may also regulate the TLR4/MD2/MyD88 complex-mediated NF-κB signaling pathway to participate in colorectal cancer cell EMT [112]. Furthermore, miR-218, miR-330-3p, and miR-498 may regulate the migration and invasion of cancer cells through the Bmi-1/Akt axis [112].

### 4.5. Bmi-1 in Cancer DNA Damage Response

An underlying feature of cancer is genomic instability, which is associated with the accumulation of DNA damage. Bmi-1 plays an important role in the regulation of DNA damage response (DDR) [113,114]. The lack of Bmi-1 in cells could cause mitochondrial dysfunction, and at the same time, the DDR pathway could be initiated with the increase of ROS [115]. Bmi-1 enhances the repair of damaged DNA through epigenetic mechanisms to reduce the genotoxic effects of ionizing radiation (IR) [116]. Bmi-1 is a component of PRC1, and affects the expression of many genes through histone H2A ubiquitination. Mechanically, Bmi-1 binds to ring2/ring1b subunit to form a functional E3 ubiquitin ligase, and inhibits the expression of multiple gene through monoubiquitination of histone H2A in lysine 119 [117]. Through Bmi-1/RIN1b E3 ubiquitin ligase, Bmi-1 promotes histone H2A and γH2AX ubiquitination for the repair of double-stranded DNA breaks by stimulating homologous recombination and non-homologous end connection [117,118]. In addition, the knockout of Bmi-1 further aggravated the phosphorylation of checkpoint kinase 2 (Chk2) and H2AX induced by cisplatin treatment [119].

### 4.6. Bmi-1 in Cancer Inflammation

Inflammation is often associated with the progression of cancer. Cells responsible for cancer-related inflammation are genetically stable and therefore do not rapidly emerge drug resistance. Therefore, targeting inflammation is an attractive strategy for both cancer prevention and cancer therapy [120]. NF-κB transcription factors are major mediators of inflammatory processes and key players in innate and adaptive immune responses. Therefore, NF-κB pathway has been considered as the target of anti-inflammatory drugs. Recent studies have found that Bmi-1 is involved in the regulation of NF-κB signaling pathway. For example, Bmi-1 promotes tumor proliferation, metastasis and drug resistance by activating NF-κB signaling pathway [121]. Similarly, Bmi-1 promotes glioma invasion by activating NF-κB/MMP3 or NF-κB/MMP9 signaling pathways [111,122]. Moreover, Bmi-1 promote colorectal cancer migration and EMT in an inflammatory microenvironment by regulating TLR4/MD2/MyD88 complex-mediated NF-κB signaling pathway [112]. Furthermore, the expression of Bmi-1 can activate the NF-κB signaling pathway and increase the expression of vascular endothelial growth factor C (VEGF-C)to promote glioma angiogenesis [123]. Additionally, toll-like receptor 4 (TLR4) promotes inflammation by inhibiting Bmi-1 to activate the NOD-like receptors (NLRs) family member (NLRP3) pathway [124].

### 4.7. Bmi-1 in Cancer Stem Cells

Stem cells have self-renewal ability, and produce at least one highly differentiated cell. Cancer stem cells refer to cancer cells with stem cell properties such as self-renewal and multicellular differentiation. At present, many studies have proved that CSCs are present in a variety of tumors, such as leukemia, breast cancer, lung cancer, glioblastoma, colon cancer, liver cancer, and others [125]. Increasing evidence has indicated that Bmi-1 plays a critical role in the self-renewal and differentiation of cancer stem cells [6,126].

Revealing the molecular mechanism of the self-renewal of neural stem cells (NSCs) is one of the main goals for understanding the homeostasis of the adult brain. Neural stem cells and progenitor cells (NSPC) strictly regulate self-renewal potential to maintain homeostasis in the brain. The researchers found that the absence of Bmi-1 led to the decline of self-renewal ability in NSCs [127,128]. Molofsky et al. further found that Bmi-1 promoted the self-renewal of NSCs by inhibiting INK4a/ARF1. One study found that Bmi-1-knockout mice lacked stem cells and developed defects in the central nervous system, but no significant effect on the proliferation of progenitor cells was recorded. The results showed that Bmi-1 is a key factor for maintaining the stability of NSCs, but had little influence on differentiated cells [129,130]. Furthermore, Protein S (PROS1) can affect the self-renewal of adult hippocampal NSPC by down-regulating Bmi-1 [131]. In summary, Bmi-1 is a vital factor for the self-renewal of NSCs.

Bmi-1 also plays a pivotal role in hematopoietic stem cells (HSCs). Park et al. reported that Bmi-1 is strongly expressed in fetal mouse and adult human HSCs. Thus, Bmi-1 knockout resulted in the reduction of proliferation of mouse bone marrow, resulting in the death of most of the studied mice before adulthood [5]. Lessard et al. [132] found that the number of stem cells in peripheral leukemia cells from Bmi-1 wild-type mice was significantly higher than that in Bmi-1 knockout mice. Subsequently, they found a decrease of self-renewal ability of HSCs and poor hematopoiesis in Bmi-1^−/−^ mice. Mice were then injected with fetal liver HSCs (Bmi-1^+/+^ or Bmi-1^−/−^) after high doses of lethal radiation. The results showed that the hematopoietic ability of bone marrow was dependent on the expression of Bmi-1.

In breast cancer stem cells (BCSCs), Bmi-1 is a target of the miR-200 family and miR-128 family. Furthermore, Bmi-1 is also regulated by certain signaling pathways, such as the Hedgehog and Wnt pathways. The Hedgehog signaling pathway promotes self-renewal in mammary stem cells and BCSCs by up-regulating Bmi-1 [4]. Bmi-1 can activate the Wnt signaling pathway by inhibiting Dickkopfs (DKK), a Wnt inhibitor gene. DKK1 resulted in the up-regulation of c-MYC, which further contributed to the transcriptional self-activation of Bmi-1 [133].

Bmi-1 is a regulator of self-renewal in prostate stem cells and a marker in intestinal stem cells [134,135]. Bmi-1 is also expressed in mesenchymal stem cells isolated from the umbilical cord [136]. Therefore, further regulatory mechanisms of Bmi-1-promoting CSC need to be revealed, and its role in tumor promotion requires exploration, to provide new therapeutic strategies for anti-tumor therapy.

### 4.8. Bmi-1 in Tumor Microenvironment

The tumor microenvironment (TME) comprises various cell types (endothelial cells, fibroblasts, immune cells, etc.) and extra-cellular components (cytokines, growth factors, hormones, extracellular matrix, etc.) that surround tumor cells and are nourished by a vascular network. Tumor cells interact with the surrounding cells through the circulatory and lymphatic systems to influence the progression of cancer and therapeutic efficacy of treatments [137,138]. The expression of Bmi-1 can also affect the tumor microenvironment (TME). Selective Bmi-1 inhibitor PTC-209 increases the expression of DKK1 by down-regulating Bmi-1, impairing in vitro osteoclast formation and destroying the tumor microenvironment [139]. There is a report that Bmi-1 is upregulated in multiple myeloma-associated macrophages (MM-MΦs) and that Bmi-1 modulates MM-MΦ’s pro-myeloma functions. Bmi-1 inhibitors could not only target multiple myeloma (MM) cells, but also eliminate MM-MΦs in the treatment of myeloma [140]. Another report suggests that tumor-associated macrophages (TAMs) may cause increased Bmi-1 expression through miR-30e* suppression, leading to gastrointestinal cancer progression [141]. At present, there are few reports on the effect of Bmi-1 on the tumor microenvironment. Therefore, the role of Bmi-1 in the tumor microenvironment needs to be further explored.

## 5. Clinical Characteristics and Cancer Therapy of Bmi-1

### 5.1. Protein Expression and Clinical Characteristics of Bmi-1

In the past two decades, many researchers have discussed the relationship between the expression level of Bmi-1 and the clinical characteristics of different cancers. Although a few results are contradictory, most of the results prove the importance of Bmi-1 in the occurrence and development of cancer. In order to describe it more simply and intuitively, we summarized the expression level and clinical characteristics of Bmi-1 in Table 2.

### 5.2. Bmi-1 in Chemoresistance and Cancer Therapy

Radiation and chemotherapy are common treatments for cancer. Drug resistance is one of the main obstacles to effective anticancer treatment. Many clinical data have reported that Bmi-1 is related to the drug resistance of tumors [114,164,165,166,167]. Silencing Bmi-1 could enhance camptothecin-induced DNA double-strand breaks and promote camptothecin-induced apoptosis. Conversely, increasing Bmi-1 can significantly reduce camptothecin-induced apoptosis [168]. Direct inhibition of Bmi-1 abrogates head and neck cancer stem-cell self-renewal and increases tumor sensitivity to cisplatin [169]. Hypoxic exposure regulates PI3K/Akt signal and EMT through activation of the HIF-1α/Bmi-1 signal to induce LSCs drug resistance [170]. Similarly, Bmi-1 activates the PI3K/Akt pathway and NF–κB pathway to promote cell growth and resistance to cisplatin treatment [170]. Furthermore, the low expression of Survivin is considered to be one of the factors for the success of chemotherapy; Bmi-1 promotes the drug resistance of B-cell lymphoma cells through the regulation of Survivin [171]. Although radiation therapy is one of the main methods of cancer treatment, tumors often acquire radioresistance, which leads to radiotherapy failure. Bmi-1 depletion makes radiation-resistant ESCC cells sensitive to radiotherapy by inducing apoptosis, senescence, ROS production, and oxidase gene expression (*Lpo*, *Noxo1* and *Alox15*), reducing DNA repair capabilities [172]. Interestingly, the inhibition of Bmi-1 activates immune responses in tumor cells and recruits CD8+ T cells. Mechanistically, knockdown of Bmi-1 not only eliminated CSCs, but also sensitized tumor cells to anti-PD-1 antibodies by recruiting CD8+ T cells [173]. In conclusion, Bmi-1 has development potential as a target for the treatment of cancer. Perhaps in future the development of small-molecule compounds that effectively target Bmi-1 will be an option for anticancer drug development.

At present, certain small molecule inhibitors of Bmi-1 have been studied, and the most commonly used small molecule inhibitor of Bmi-1 is PTC-209. An article published in 2013 reported that researchers had identified a low molecular compound PTC-209 for the first time. They also found that PTC-209 reduced the tumorigenic activity and volume of xenograft tumors and the number of functional colorectal cancer-initiating cells in mice, after short-term treatment [174]. Subsequently, the drug has also been studied in other cancer cells and has demonstrated its anti-tumor effect, including for head neck squamous cell carcinoma (HNSCC) [175], prostate cancer [176], myeloma [177], glioblastoma [178], and cervical cancer cells [179]. In 2019, researchers developed an orally active nanoparticle PTC-209 vector, which can significantly inhibit colony formation and migration of colon cancer cells, and inhibit tumor progression and metastasis in orthotopic tumor-bearing mice by reversing stemness [180]. An orally active and selective Bmi-1 inhibitor, PTC-596, downregulated bcl2 family apoptosis regulator Mcl-1 and induces p53-independent mitochondrial apoptosis in AML progenitor cells [181]. The drug has also been explored for treatment of other cancers, for example, mantle cell lymphoma [182], cancer stem-like cells [99], multiple myeloma [183], and glioblastoma multiforme [184]. PTC-596 has been used in clinical phase 1 for conditions including diffuse intrinsic pontine glioma and leiomyosarcoma [185,186]. Moreover, a novel inhibitor, RU-A1, was developed based on the RNA three-dimensional (3D) structure of Bmi-1 and showed stronger potency than PTC-209 in targeting CSCs [187]. PTC-028, an orally bioavailable compound, mediates hyperphosphorylation of Bmi-1 accompanied by a transient decrease in ATP and disruption of mitochondrial redox balance, enhancing caspase-dependent apoptosis. In an orthotopic mouse model of ovarian cancer, oral administration of PTC-028 as a single agent demonstrated significant antitumor activity comparable to standard cisplatin/paclitaxel therapy [188]. Researchers have also discovered another small molecule compound QW24, which can inhibit the expression of Bmi-1 through the lysosomal autophagy pathway, and can inhibit the growth of mouse xenograft tumors and liver metastasis, and prolong the survival period of mice [189]. Unfortunately, there have been no other reports of the anticancer effects of QW24, so its anti-cancer effect needs to be further verified. Table 3 summarizes the inhibitors of Bmi-1, the ways they regulate Bmi-1, and the regulatory mechanisms in tumors.

## 6. Future Research and Conclusions

PcG family member Bmi-1 plays a vital role in the proliferation, apoptosis, metastasis, and chemical sensitivity of cancer cells. The discovery of the biogenesis and function of non-coding RNA has improved our understanding of the complexity of the human genome. With the deepening of research, it has been found that non-coding RNA is involved in the regulation of the cell cycle, proliferation, and differentiation, especially in the occurrence and development of cancer. Many miRNAs inhibit Bmi-1 expression by targeting the 3′UTR [22]. That means that these miRNAs can also affect cancer proliferation, migration, invasion, apoptosis, and drug sensitivity [22]. Thus, the interaction between miRNA and Bmi-1 plays an important role in the occurrence and development of cancer. This also provides new strategies for cancer treatment. However, at present, the research on the interaction between miRNA and Bmi-1 remains in the initial stages, and there are many problems still to be solved.

As a proto-oncogene, Bmi-1 has been confirmed to be highly expressed in a variety of tumors. It is related to the clinical stage, pathological classification, and lymph node metastasis, and can be used as one of the predictors for prognosis and recurrence in tumor patients. With the development of molecular biological technology and the application of gene chip technology, development is expected of a microfluidic multi-indicator joint inspection chip, a circulating cancer-cell capture chip, and other devices for automatic detection of Bmi-1 expression [208,209,210]. In addition, Bmi-1 is involved in the occurrence and development of many tumors, and the targeted therapy of cancer stem cells is an important measure for future development. However, there is currently no drug that can be used clinically to target Bmi-1 specifically. Therefore, the discovery of new small-molecule compounds that specifically inhibit Bmi-1 will be the focus of future research.

## Figures and Tables

**Figure 1 ijms-23-08231-f001:**
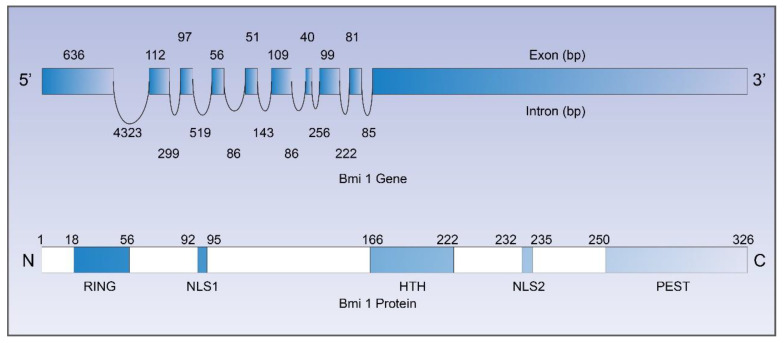
The structure of gene and protein of Bmi-1. The *Bmi-1* gene contains 10 exons and 9 introns. The amino acid sequence of Bmi-1 protein contains a RING finger domain, a helix–turn–helix, two nuclear localization signals (NLS), and a PEST region.

**Figure 2 ijms-23-08231-f002:**
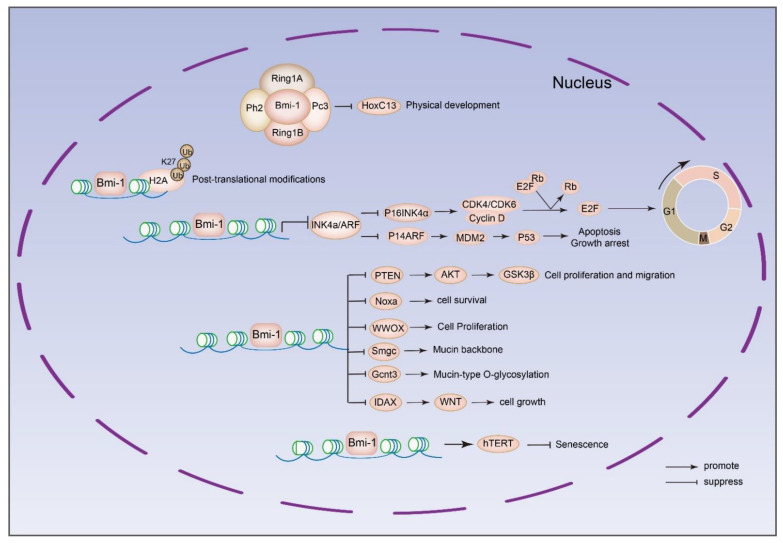
Downstream genes directly regulated by Bmi-1. Bmi-1 can enhance H2A ubiquitination and bind to the promoter of *HoxC13* to reduce the expression of *HoxC13* in HeLa cells [77]. The most classic downstream target of Bmi-1 is INK4a/ARF. The tumor suppressor gene INK4a/ARF can encode two regulatory genes: *p16^INK4a^* and *p14^ARF^* (*p19^ARF^* in mice). P16^INK4a^ arrests cells in G0/G1 phase through cyclinD-CDK4/6-pRb-E2F. p14^ARF^ regulates the cell apoptosis through MDM2-p53 pathway [76]. Bmi-1 can regulate the development of colon cancer by negatively regulating the expression level of *PTEN* and then activating the Akt/GSK3β axis [78]. Bmi-1 regulates memory CD4 T cell survival via repression of the proapoptotic BH3-only protein *Noxa* gene [80]. Bmi-1 regulates cell fate via transcriptional repression tumor suppressor WW Domain Containing Oxidoreductase (WWOX) in small-cell lung cancer cells [81]. Bmi-1 suppresses the expression of Smgc gene and Gcnt3 gene [82]. Bmi-1 activates Wnt signaling in colon cancer by negatively regulating the Wnt antagonist *IDAX* [83]. Bmi-1 can also transcriptionally positively regulate human telomerase reverse transcriptase (hTERT) to block senescence in human mammary epithelial cells [79].

**Table 1 ijms-23-08231-t001:** MicroRNAs that inhibit Bmi-1 in tumors.

miRNAs	Efficacy for Cancer	Cancer Type	References
miR-34a, miR-15a, miR-218, miR-183, miR-498, miR-128	Inhibits proliferation; inhibits metastasis; decreases chemoresistance	Gastric cancer	[23,24,25,26,27]
miR-218, miR-200c, miR-485-5p	Inhibits proliferation; inhibits migration; increases apoptosis	Colorectal cancer	[28,29,30]
miR-15a, miR-183	Inhibits proliferation and EMT	Pancreatic ductal adenocarcinoma	[31,32]
miR-203	Inhibits self-renewal	Esophageal cancer	[33]
miR-203	Promotes apoptosis	Oral cancer	[34]
miR-218, miR-203	Inhibits proliferation and invasion; introduces apoptosis; decreases radiosensitivity and chemosensitivity	Hepatocellular carcinoma	[35,36,37,38,39]
miR-218	Increases chemosensitivity	Liver cancer	[38]
miR-320a	Inhibits proliferation and migration	Nosopharyngeal carcinoma	[40]
miR-132, miR-498	Increases radiosensitivity; inhibits proliferation and invasion	Cervical cancer	[41,42]
miR-128, miR-200b, miR-221, miR-30d, miR-15a, miR-330-3p, miR-212	Inhibits proliferation and migration; increases chemosensitivity	Prostate cancer	[43,44,45,46,47,48,49,50]
miR-200c, miR-194	Inhibits proliferation; inhibits EMT	Endometrial carcinoma	[51,52]
miR-128, miR-495	Inhibits proliferation; introduces apoptosis; increases chemosensitivity and DNA damage	Breast cancer	[53,54]
miR-132, miR-15a, miR-16, miR-128	Inhibits metastasis; increases chemosensitivity	Ovarian cancer	[55,56,57]
miR-361-5p, miR-218, miR-128	Inhibits proliferation; inhibits EMT	Glioma	[58,59,60]
miR-128, miR-16, miR-128a	Inhibits proliferation and angiogenesis; introduces radiosensitivity	Glioblastoma	[61,62,63]
miR-128a	Inhibits ROS	Medulloblastoma	[64]
miR-218	Inhibits proliferation, inhibits migration, inhibits apoptosis	Osteosarcoma	[65,66]
miR-200c, miR-139-5p, miR-218, miR-15	Inhibits proliferation; inhibits metastasis; inhibits apoptosis; inhibits autophagy	Bladder cancer	[13,67,68,69]
miR-218	Inhibits proliferation	Acute Promyelocytic Leukemia	[70]
miR-218	Inhibits proliferation and metastasis	Lung adenocarcinoma	[71]
miR-203	Inhibits proliferation	Myeloma	[72]
miR-200C	Inhibits proliferation and metastasis	Renal cancer	[73]
miR-154	Inhibits proliferation and migration	Non-small cell lung cancer	[74]
miR-200c	Inhibits proliferation and migration; increases chemosensitivity	Melanoma	[75]

**Table 2 ijms-23-08231-t002:** Protein expression and clinical characteristics of Bmi-1.

Cancer Type	mRNA/Protein	High/Low Expression	Positive Percentage	Clinical Characteristics	Remarks	Ref.
Gastric cancer	Protein	High	GC162 (52.5%)	Associated with Lauren’s and Borrmann’s classification and clinical stage	Mainly in nucleus	[142]
	Protein	High	GC 178 (70.8%)	Associated with sex, gross type, and histologic type	Mainly in nucleus	[143]
	mRNA	High	71	Associated with tumor size, depth of invasion, lymph node metastasis, and clinical stage	Not involved	[144]
Nonsmall cell lung cancer	Protein and mRNA	High		Associated with tumor size, poor differentiation, and distant metastasis	Mainly in nucleus	[108]
Endometrial Carcinoma	Protein	High	48	A significant positive relationship between Bmi-1 and Ki-67, cyclin A, or p53	Mainly in nucleus	[145]
Esophageal cancer	Protein	High	1523	Associated with differentiation, tumor/node/metastasis stage, depth of invasion, and lymph node metastasis	Mainly in nucleus	[146]
Cervical cancer	Protein and mRNA	High	302 (55.3%)	Correlated with clinical stage, lymph node metastasis, vascular invasion, and human papillomavirus (HPV) infection	Mainly in nucleus	[147]
Acute myeloid leukemia (AML)	mRNA	High	60	Showed a strong association with failure to achieve complete remission (CR) or with relapse	Not involved	[148]
Esophageal squamous cell carcinoma (ESCC)	Protein	High	80 (78.7%)	Correlated with depth of invasion and lymph node metastasis, but not with patient age, tumor size, or nationality	Not involved	[149]
ESCC	Protein and mRNA	High	171 (64.3%)	Correlated with stage and pN classification.	Mainly in nucleus	[150]
Endometrial adenocarcinoma	Protein	High	60	Correlated with FIGO stage, myometrial invasion, and lymph node metastasis	both the nucleus and cytoplasm	[106]
Colon cancer	Protein and mRNA	High	203 (66.5%)	Correlated with clinical stage, depth of invasion, nodal involvement, distant metastasis, and Ki67 level	Mainly in nucleus	[151]
Uterine cervical cancer	Protein	High	152	Correlated with tumor size, clinical stage, and regional lymph nodes metastasis	Mainly in nucleus	[152]
Bladder cancer	Protein	High	137	Correlated withhistopathological classification, clinical stage, recurrence, and patient survival	Mainly in nucleus	[153]
Ovarian carcinoma	Protein	Low	179	Correlated with tumors’ histological type, grade, pT/pN/pM status, and FIGO stage	both the nucleus and cytoplasm	[154]
Epithelial ovarian cancer	Protein	High	40 (72.5%)	Associated with advanced International Federation of Gynecology and Obstetrics stages, bilaterality, and higher Gynecologic Oncology Group grades and carcinomas of serous histology	Mainly in nucleus	[155]
Uterine cervical cancer	mRNA	High	109	Correlated with clinical stage and lymph nodes metastasis	Not involved	[156]
Salivary adenoid cystic carcinoma	Protein	High	10	Associated with tumor metastasis, Snail, Slug, and E-cadherin, serves as a highrisk for *AdCC*	Not involved	[157]
Laryngeal carcinoma	Protein	High	64 (84.4%)	Correlated with distant metastasis, N pathological status, T classification	B oth the nucleus and cytoplasm	[158]
Pancreatic cancer	Protein	High	72 (48.61%)	Correlated with the presence of lymph node metastases and negatively correlated with patient survival rates	Mainly in nucleus	[159]
Squamous cell carcinoma of the tongue	Protein	High	73 (82%)	Correlated with recurrence	in nucleus	[160]
ovarian Carcinoma	mRNA	High	47(72.34%)	Correlated with tumor grade	both the nucleus and cytoplasm	[161]
Neuroblastoma	Protein	High	45	Correlated with MYCN	in nucleus	[162]
Pediatric brain tumors	mRNA	High	56	Expression of Bmi-1 showed significant differences between high-grade tumors and low-grade tumors	Not involved	[163]

**Table 3 ijms-23-08231-t003:** Inhibitors of Bmi-1.

Inhibitor Name	Regulatory for Bmi-1	Tumor Type	Regulatory Mechanism	References
PTC-209	Reduces transcript levels	Cervical cancer	Promotes cell G0/G1 arrest and apoptosis	[179]
		Colon cancer	Developed an orally active, easily synthesized PTC209 nanomedicine	[180]
		Alveolar rhabdomyosarcoma	Activates the Hippo pathway	[190]
		Glioblastoma	Inhibits glioblastoma cell proliferation and migration	[178]
		Ovarian cancer	Induces autophagy through ATP depletion	[101]
		Lung cancer cells, breast cancer cells and colon cancer cells	Inhibits STAT3 Phosphorylation	[191]
		Pluripotent stem cells	Reduces the expression of neuronal markers, such as Nestin	[192]
		Prostate cancer	Efficiently targets Bmi-1 and Sox2	[176]
		ESCC	Inhibits ESCC progression when combined with cisplatin	[193]
		Acute myeloid leukemia	Inhibits proliferation and induce apoptosis	[194]
		Acute Leukemia Cells	Down-regulates the expression of Notch signaling proteins Notch1, Hes1, and MYC	[195]
		Acute myeloid leukemia	Reduces protein level of Bmi-1 and its downstream target mono-ubiquitinated histone H2A and induces apoptosis	[196]
		Breast cancer	Transcriptionally upregulates expression of miR-200c/141 cluster	[197]
		Biliary tract cancer cells	Causes down-regulation of cell cycle-promoting genes, DNA synthesis gene and DNA repair gene	[198]
		Chronic myeloid leukemia cells	Triggers CCNG2 expression	[105]
		MM	Down-regulates the expression of Bmi-1 protein and the associated repressive histone mark H2AK119ub	[177]
		HNSCC	Inhibits proliferation, migration and invasiveness, increases cell apoptosis and chemosensitivity	[175]
PTC596	Reduces protein levels of BMI-1	Myeloma	Induces cell cycle arrest at G2/M phase followed by apoptotic cell death	[199]
		AML	Downregulates Mcl-1 and induces p53-independent mitochondrial apoptosis	[181]
		Glioblastoma	Targets both Bmi-1 and EZH2, prevents GBM colony growth and CSC self-renewal	[184]
		Mantle cell lymphoma	Induces mitochondrial apoptosis, loss of mitochondrial membrane potential, C-caspase-3, Bax activation, and phosphatidylserine externalization	[182]
		Cancer stem-like cells	Induces apoptosis through DUB3-mediated Mcl-1 degradation	[99]
		Pancreatic ductal adenocarcinoma (PDA)	Induces mitotic arrest and apoptosis	[200]
		Diffuse intrinsic pontine glioma (DIPG)	Decreases tumor volume and growth kinetics, increases intertumoral apoptosis, and sustains animal survival benefit.	[201]
RU-A1	Bind to the Bmi-1 mRNA	Hepatocellular carcinoma	Impairs cell viability, reduces cell migration, enhances HCC cell sensitivity to 5-fluorouracil (5-FU) in vitro	[187]
PTC-028	Posttranslational modification	multiple myeloma	Impairs MYC and Akt signalling activity; induces cell cycle arrest at G2/M phase and apoptotic	[183]
		Diffuse intrinsic pontine glioma (DIPG)	Decreases the expression of E2F1, KRAS, Nestin, SOX2 while increases the expression of p21 and differentiation markers (GFAP)	[202]
	Hyperphosphorylation	Ovarian cancer	Decreases ATP and a compromised mitochondrial redox balance potentiate caspase-dependent apoptosis	[188]
		Medulloblastoma (MB)	Abolishes the self-renewal capacity of MB stem cells, reduces tumor initiation ability of recurrent MB cells	[203]
		Endometrial cancer	Reduces cell invasive capacity and enhances caspase-dependent apoptosis	[204]
		Alveolar rhabdomyosarcoma	Inhibits proliferation and causes tumor growth delay in vivo	[190]
QW24	Autophagy-lysosome degradation pathway	Stem-like colorectal cancer	Inhibits self-renewal of colorectal cancer-initiating cells (CICs)	[189]
SH498		Colorectal cancer	Reduces PRC1 complex activity by down-regulating Bmi-1 and ub-H2A	[205]
Artemisinin	Protein and transcript levels	Nasopharyngeal carcinoma	Induces G1 cell cycle arrest via the Bmi-1-p16/CDK4 axis	[206]
PRT4165	Down-regulating Bmi-1/RING1A self-ubiquitination	Acute leukemia	Increases cell apoptosis	[195]
CDDO-Me		ESCC	Induces autophagy via suppression of PI3K/Akt/mTOR signaling pathway	[207]

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
