# Peer review of "The Crucial Roles of Bmi-1 in Cancer: Implications in Pathogenesis, Metastasis, Drug Resistance, and Targeted Therapies"

_ijms, 2022, doi:10.3390/ijms23158231_

Round 1

Reviewer 1 Report

Comments for Authors

Dear authors,

Your Review will be useful to clinicians and practicing oncologists, but needs to be seriously revised.

General recommendations:

1. The separation of chapter 4 is not justified. Right now it looks like a list of poorly organised and unconnected facts. It would be better if you integrated it piecemeal into chapter 5, which should be renamed.

2. Chapter 6 consists mainly of a table and it is strange. It would be better if this chapter became part of chapter 7, which discusses Bmi-1-oriented cancer therapy 

3. All genes and proteins that are mentioned in the review should be briefly characterised and not just mentioned, as the review will be read by clinicians, molecular biologists and biochemists with different backgrounds.

4. If your illustrations are based on drawings from other articles, you should refer to them and possibly request permission to use them directly

5. All abbreviations of gene and protein names in the text should be deciphered. The abbreviations can be given at the end of the article, in the Abbreviations section

6. Please check grammar and spelling. Remember that the names of genes are written in italics. There are many mistakes in the text and only a few of them I have noted, as my work is not editorial

Listed below are detailed comments on your text. Please pay close attention to them.

Abstract:

1.      In fact, Bmi-1 has multiple functions in cancer biology and is closely related to many classical molecules

It is not clear which molecules you are talking about

Introduction

2.      PcG proteins act on the development of organisms by forming two polymer complexes

multimeric protein complexes

3.      Over the years, it has been identified that MicroRNAs (miRNAs) play important roles in various biological behaviors and processes by inhibiting the translation of mRNAs and inducing their degradation, which makes them potential molecular targets for cancer therapy

What is meant by "behaviours"? Better to remove it and leave "biological processes by..."

4.      Therefore, in this review, we first describe the biological function of Bmi-1. Then, we reviewed the relevant signaling pathways regulated by Bmi-1 in biology and carcinogenesis and its regulatory network with miRNAs. To provide a reference for a more comprehensive understanding of the role of Bmi-1 in cancer.

Carcinogenesis is also a biological process. Replace "biological function of Bmi-1" and "Bmi-1 in biology and carcinogenesis" with "normal function of Bmi-1" and "Bmi-1 in normal and carcinogenesis".

Molecular features and characteristics of Bmi-1

5.      The amino acid sequence of Bmi-1 protein contains several significant motifs, a RING finger domain, a helix-turn-helix-turn-helix-turn, two nuclear localization signals (NLSs), and a PEST region.

Misprint. The motif is called helix-turn-helix (HTH)

6.      The RING finger domain is in the N-terminal and is composed  of a C3HC4 conserved sequence and zinc finger

The C3HC4 sequence itself (Cys-X2-Cys-X9-39-Cys-X1-3-His-X2-3-Cys/His-X2-Cys- X4-48-Cys-X2-Cys, where X can be any amino acid) forms the finger domain in RING protein. Your text implies that these are different motifs.

7.      Besides, the PEST region is situated at the C terminal, containing more serine, glutamic acid, threonine, and proline residues[15], which is associated with turnover of intracellular Bmi-1 protein

Besides, the PEST region situated at the C terminal, contain many serine, glutamic acid, threonine, and proline residues[15], which are associated with intracellular turnover of  Bmi-1 protein.

8.      (Figure. 1) illustrates the structure of the Bmi-1 gene and Bmi-1 protein[16].

The Figure 1 illustrates…

9.      Figure 1.

You need to correct the mistake in the name of the HTH motif and be sure to provide a link to the article from which the illustration is redrawn - (with modifications from Sahasrabuddhe, 2016).

Upstream regulatory mechanisms of Bmi-1

10.  Over the past decade, researchers have attempted to understand the regulation of Bmi-1 at the transcriptional, post-transcriptional and post-translational levels, but research progress has been modest.

It is not entirely clear in which area progress has been modest. From the text below, it appears that many transcription factors have been found that positively and negatively regulate Bmi-1 transcription. The author of the review to which you are referring lists these factors with references to 12 studies. 

11.  Most proteins can undergo PTMs, and there are more than 200 known covalent modifications of proteins, including phosphorylation,  nitrosylation, nitration, ubiquitination, and small ubiquitin-related modifiers (SUMO), etc.

The sentence is inconsistent - first you talk about modifications, but then you list modifier proteins in the same row as them.

12.  Bmi-1 represses the expression of the Ink4a-Arf locus, which encodes two tumor suppressors (p16Ink4a and p19Arf) that are key regulators of senescence. Akt phosphorylates Bmi-1 at Ser 316, which leads to the dissociation of Bmi- 1 from chromatin and represses of the Ink4a-Arf locus

The text does not explain what Akt is.

Two sentences contradict each other. The first one says that Bmi-1 represses the Ink4a-Arf locus, but the second one implies that dissociation of Bmi- 1 from chromatin leads to repression of the Ink4a-Arf locus.

13.  In addition, Bmi-1 can also be sumoylated. The study found that DNA damage can induce the sumoylation of lysine 88 of Bmi-1, and knockout of CBX4 can completely eliminate this phenomenon, which means that CBX4 can mediate the sumoylation of Bmi-1

Need to clarify what this protein modification is and what CBX4 is

14.  Furthermore, Ser255 be found to be the site of O-glcn acylation of Bmi-1 in prostate cancer, and O-glcn acylation of Bmi-1 promote the stability of Bmi-1 protein and its oncogenic activity

A misprint. O-GlcNAcylation

15.  MiRNAs is a small class of ncRNA that contains about 19-24 nt.

 MicroRNAs (miRNAs or miRs) are a class of small  of ncRNAs that contains about 19-24 nt.

16.  Research has found that miRNAs mainly target two spacers of the 3'UTR of Bmi-1. The first target position is 469-725, such as miR-128, miR-221, miR-183, miR-200b/c; The   other position is 1442-1758, such as miR-203, miR-218.

Research has found that miRNAs mainly target two spacers of the 3'UTR of Bmi-1. The first target position is 469-725, for such as miR-128, miR-221, miR-183, miR-200b/c; The other position is 1442-1758, for such as miR-203, miR-218.

17.  Table 1.

The title of the first column is 'Regulatory Factor', but it only refers to miRs.

Some miRs are capitalised for some reason.

Downstream signaling pathways mediated by Bmi-1

Bmi-1 transcriptionally regulates downstream genes

18.  Figure 2.

The sequence of descriptions should follow the order in which the material is presented in the illustration

19.  mTOR signaling pathway

Sometimes large letters are used unnecessarily in chemical names.

There is no reference to an illustration (Figure 3.).

20.  Figure 3.

C and D have to be swapped out.

NF-κB signaling pathway

21.  Nuclear factor-κB (NF-κB) transcription factors are master regulators of inflammation and immune homeostasis.

Transcription factors belonging to the family Nuclear Factor-κappa B (NF-κB) are master regulators of inflammation and immune homeostasis.

22.  (Fig.3B) briefly describes the crosstalk relationship between Bmi-1 and NF-κB signaling pathway.

The crosstalk  between Bmi-1 and NF-κB signaling pathway briefly describes on Figure 3B.

23.   Inhibition of Bmi-1 inhibits osteosarcoma cell growth by inhibiting the expression of MMP9 and NF-κB signaling

Use synonyms

24.  Other signaling pathway

Other signaling pathways

25.  Figure 3D.

The interactions of Bmi-1, ink4a-ARF and c-MYC are not reflected in Figure 3D, while the other pathways shown in the picture are not described in the text.

26.   Notably, studies have shown that expression levels of Bmi-1 can contribute to the stabilization of YAP by blocking YAP in the nucleus

It is not clear from the text whether low or high levels of Bmi-1 expression stabilise YAP.

27.  Bmi-1 in cancer

This is an unfortunate title for a chapter, as the whole article is about Bmi-1 in cancer. You need to clarify the subject of the chapter

28.  The function of Bmi-1 in tumors has been identified in various pathological pro- cesses, including abnormal cell proliferation, evasion of apoptosis, tumor migration, and  stemness maintenance of cancer stem cells (CSCs)

migration of cancer cells

Bmi-1 in cancer proliferation

29.  Mechanismly, when Bmi-1 is deficient, p16Ink4a is up-regulated, which prevents the binding of CDK4/6 to cyclinD, leading to phosphorylation of Rb[92]. Phosphorylated Rb binds to E2F to inhibit E2F-mediated tran- scription, which arrests the cell cycle in G0/G1 phase[102].

References 92 and 102 are mixed up.

30.  For  example, Bmi-1-shRNA reduces the expression of cyclin D1 and increases the expression of p21/p27 through the INK4a-p16 independent pathway to arrest lung adenocarcinoma cells in G0/G1 phase[107].

There is no example in this sentence to illustrate the above

31.  Interestingly, Bmi-1 can interact with c-MYC to coordinately regulate progression of  cell cycle and promote tumor formation[113]

If you are talking about protein-protein interactions between Bmi-1 and c-MYC and not about the regulation of transcription as discussed in the previous chapters, this needs to be dealt with separately

32.  Bmi-1 in cancer apoptosis

It is very difficult to draw conclusions from this chapter - which abnormalities in Bmi-1, i.e. an increase or decrease in expression, lead to a disruption of apoptosis.  If the different tumour types differ in this parameter - you need to list the variants so that practitioners can take advantage of this

Bmi-1 in cancer DNA damage response

33.  The lack of Bmi-1 in cells will cause mitochondrial dys- function, and at the same time, the DDR pathway will be initiated with the increase of ROS

Is the use of the future tense justified?

34.  Mechanically, Bmi-1 binds to ring2/ring1b subunit to form a functional E3 ubiquitin ligase, and inhibits the expression of multiple gene through monoubiquitination of histone H2A in lysine 119 and Lys120 (Lys118 and Lys119 in H2A)[129]

Which lysines are modified?

Bmi-1 in cancer stem cells

35.  Stem cell has self-renewal ability, and produce at least one highly differentiated cell.  Cancer stem cells (CSCs) refer to cancer cells with stem cell properties.

What properties are you talking about? You need to be more specific.

36.  5.7 Others

Unfortunate title of the chapter. "Others" what?

37.  PTC- 209 increases the expression of DKK1 by down-regulating Bmi-1, impairs in vitro osteoclast formation and destroys the tumor microenvironment

Selective BMI-1 inhibitor PTC- 209 increases the expression of DKK1 by down-regulating Bmi-1, impairs in vitro osteo- clast formation and destroys the tumor microenvironment

38.  6.Clinical characteristics and cancer therapy of Bmi-1

6.1Protein expression and clinical characteristics of Bmi-1

Where is 6.2?

It is not a good idea to have the chapter as a table. It is better if the table and the small accompanying text form part of the next chapter, the title of which should be changed to reflect this part.

Bmi-1 in chemoresistance and cancer Therapy

39.  Interference of Bmi-1 in cancer     cells enhances Camptothecin-induced DNA double-strand breaks and promotes Camptothecin-induced apoptosis.

What kind of interference are you talking about?

The chapter does not cover other Bmi-1 inhibitors:

PTC596 (Flamier, Anthony, et al. "Off-target effect of the BMI1 inhibitor PTC596 drives epithelial-mesenchymal transition in glioblastoma multiforme." NPJ precision oncology 4.1 (2020): 1-10.)

RU-A1 (Bartucci, Monica, et al. "Synthesis and characterization of novel BMI1 inhibitors targeting cellular self-renewal in hepatocellular carcinoma." Targeted oncology 12.4 (2017): 449-462.)

PTC-028 (Dey, Anindya, et al. "Evaluating the mechanism and therapeutic potential of PTC-028, a novel inhibitor of BMI-1 function in ovarian cancer." Molecular cancer therapeutics 17.1 (2018): 39-49.)

Future research and conclusions

40.  In recent years, PcG family member Bmi-1 plays a vital role in proliferation, apopto sis, metastasis, chemical sensitivity of cancer cells.

Why only in recent years?

41.  While MiRNA mainly inhibits the expression of Bmi-1 by targeting the 3'UTR of Bmi-1[24]. For instance, the miRNA can inhibit the expression of Bmi-1 and then affect the proliferation, migration, invasion, apoptosis, and drug sensitivity of cancer

Many miRNAs inhibit Bmi-1 expression by targeting the 3'UTR [24-80]. That means, these miRNAs can affect cancer proliferation, migration, invasion, apoptosis and drug sensitivity [26, 55, 63].

42.  As a proto-oncogene, Bmi-1 has been confirmed to be highly expressed in a variety of tumors, and it is related to the clinical stage, pathological classification, lymph node metastasis and other malignancy of tumors, and can be used as one of the predictors of prognosis and recurrence of tumor patients.

The parameters listed fall into different logical categories.

43.  With the development of molecular biology technology and the application of gene chip technology, it is expected to develop a micro- fluidic multi-indicator joint inspection chip, a circulating cancer cell capture chip, and devices for automatic detection of Bmi-1 expression.

Need links to these new technologies

Respectfully,

your reviewer

Author Response

  1. The separation of chapter 4 is not justified. Right now it looks like a list of poorly organised and unconnected facts. It would be better if you integrated it piecemeal into chapter 5, which should be renamed.

Answer: Thank you very much for your suggestion,we have reorganized this chapter and supplement accordingly in the manuscript.

  1. Chapter 6 consists mainly of a table and it is strange. It would be better if this chapter became part of chapter 7, which discusses Bmi-1-oriented cancer therapy

Answer: Thank you very much for your suggestion, we have merged the two parts.

  1. All genes and proteins that are mentioned in the review should be briefly characterised and not just mentioned, as the review will be read by clinicians, molecular biologists and biochemists with different backgrounds.

Answer: Thank you very much for your review, we have supplemented in the manuscript.

  1. If your illustrations are based on drawings from other articles, you should refer to them and possibly request permission to use them directly

Answer: Thank you very much for your review, we have supplemented in the manuscript.

  1. All abbreviations of gene and protein names in the text should be deciphered. The abbreviations can be given at the end of the article, in the Abbreviations section

Answer: Thank you very much for your review, we have supplemented in the manuscript.

  1. Please check grammar and spelling. Remember that the names of genes are written in italics. There are many mistakes in the text and only a few of them I have noted, as my work is not editorial

Listed below are detailed comments on your text. Please pay close attention to them.

Answer: Thank you very much for your review. We have made corrections in the corresponding parts of the manuscript.

Abstract:

  1. In fact, Bmi-1 has multiple functions in cancer biology and is closely related to many classical molecules

It is not clear which molecules you are talking about

Answer: Thank you very much for your review, we have supplemented in the manuscript.

Introduction

  1. PcG proteins act on the development of organisms by forming two polymer complexes

multimeric protein complexes

Answer: Thank you very much for your review. We have made corrections in the corresponding parts of the manuscript.

  1. Over the years, it has been identified that MicroRNAs (miRNAs) play important roles in various biological behaviors and processes by inhibiting the translation of mRNAs and inducing their degradation, which makes them potential molecular targets for cancer therapy

What is meant by "behaviours"? Better to remove it and leave "biological processes by..."

Answer: Thank you very much for your review. We have made corrections in the corresponding parts of the manuscript.

  1. Therefore, in this review, we first describe the biological function of Bmi-1. Then, we reviewed the relevant signaling pathways regulated by Bmi-1 in biology and carcinogenesis and its regulatory network with miRNAs. To provide a reference for a more comprehensive understanding of the role of Bmi-1 in cancer.

Carcinogenesis is also a biological process. Replace "biological function of Bmi-1" and "Bmi-1 in biology and carcinogenesis" with "normal function of Bmi-1" and "Bmi-1 in normal and carcinogenesis".

Answer: Thank you very much for your review. We have made corrections in the corresponding parts of the manuscript.

Molecular features and characteristics of Bmi-1

  1. The amino acid sequence of Bmi-1 protein contains several significant motifs, a RING finger domain, a helix-turn-helix-turn-helix-turn, two nuclear localization signals (NLSs), and a PEST

Misprint. The motif is called helix-turn-helix (HTH)

Answer: Thank you very much for your review. We have made corrections in the corresponding parts of the manuscript.

  1. The RING finger domain is in the N-terminal and is composed  of a C3HC4 conserved sequence and zinc finger

The C3HC4 sequence itself (Cys-X2-Cys-X9-39-Cys-X1-3-His-X2-3-Cys/His-X2-Cys- X4-48-Cys-X2-Cys, where X can be any amino acid) forms the finger domain in RING protein. Your text implies that these are different motifs.

Answer: Thank you very much for your review, we have changed in the manuscript.

  1. Besides, the PEST region is situated at the C terminal, containing more serine, glutamic acid, threonine, and proline residues[15], which is associated with turnover of intracellular Bmi-1 protein

Besides, the PEST region situated at the C terminal, contain many serine, glutamic acid, threonine, and proline residues[15], which are associated with intracellular turnover of  Bmi-1 protein.

Answer: Thank you very much for your review, we have changed in the manuscript.

  1. (Figure. 1) illustrates the structure of the Bmi-1 gene and Bmi-1 protein[16].

The Figure 1 illustrates…

Answer: Thank you very much for your review, we have made corrections in the corresponding parts of the manuscript.

  1. Figure 1.

You need to correct the mistake in the name of the HTH motif and be sure to provide a link to the article from which the illustration is redrawn - (with modifications from Sahasrabuddhe, 2016).

Answer: Thank you very much for your review, we have supplemented in the manuscript.

Upstream regulatory mechanisms of Bmi-1

  1. Over the past decade, researchers have attempted to understand the regulation of Bmi-1 at the transcriptional, post-transcriptional and post-translational levels, but research progress has been

It is not entirely clear in which area progress has been modest. From the text below, it appears that many transcription factors have been found that positively and negatively regulate Bmi-1 transcription. The author of the review to which you are referring lists these factors with references to 12 studies. 

Answer: Thank you very much for your review, we have changed in the manuscript.

  1. Most proteins can undergo PTMs, and there are more than 200 known covalent modifications of proteins, including phosphorylation,  nitrosylation, nitration, ubiquitination, and small ubiquitin-related modifiers (SUMO),

The sentence is inconsistent - first you talk about modifications, but then you list modifier proteins in the same row as them.

Answer: Thank you very much for your review, we have changed in the manuscript.

  1. Bmi-1 represses the expression of the Ink4a-Arf locus, which encodes two tumor suppressors (p16Ink4a and p19Arf) that are key regulators of senescence. Akt phosphorylates Bmi-1 at Ser 316, which leads to the dissociation of Bmi- 1 from chromatin and represses of the Ink4a-Arf locus

The text does not explain what Akt is.

Answer: Thank you very much for your review, we have made a description in the manuscript.

Two sentences contradict each other. The first one says that Bmi-1 represses the Ink4a-Arf locus, but the second one implies that dissociation of Bmi- 1 from chromatin leads to repression of the Ink4a-Arf locus.

Answer: Thank you very much for your review, we have changed in the manuscript.

  1. In addition, Bmi-1 can also be sumoylated. The study found that DNA damage can induce the sumoylation of lysine 88 of Bmi-1, and knockout of CBX4 can completely eliminate this phenomenon, which means that CBX4 can mediate the sumoylation of Bmi-1

Need to clarify what this protein modification is and what CBX4 is

Answer: Thank you very much for your review, we have made a detailed description in the manuscript.

  1. Furthermore, Ser255 be found to be the site of O-glcn acylation of Bmi-1 in prostate cancer, and O-glcn acylation of Bmi-1 promote the stability of Bmi-1 protein and its oncogenic activity

A misprint. O-GlcNAcylation

Answer: Thank you very much for your review, we have made corrections in the manuscript.

  1. MiRNAs is a small class of ncRNA that contains about 19-24 nt.

 MicroRNAs (miRNAs or miRs) are a class of small of ncRNAs that contains about 19-24 nt.

Answer: Thank you very much for your review, we have made corrections in the manuscript.

  1. Research has found that miRNAs mainly target two spacers of the 3'UTR of Bmi-1. The first target position is 469-725, such as miR-128, miR-221, miR-183, miR-200b/c; The other position is 1442-1758, such as miR-203, miR-218.

Research has found that miRNAs mainly target two spacers of the 3'UTR of Bmi-1. The first target position is 469-725, for such as miR-128, miR-221, miR-183, miR-200b/c; The other position is 1442-1758, for such as miR-203, miR-218.

Answer: Thank you very much for your review, we have changed in the manuscript.

  1. Table

The title of the first column is 'Regulatory Factor', but it only refers to miRs.

Some miRs are capitalised for some reason.

Answer: Thank you very much for your review, we have changed in the manuscript.

Downstream signaling pathways mediated by Bmi-1

Bmi-1 transcriptionally regulates downstream genes

  1. Figure 2.

The sequence of descriptions should follow the order in which the material is presented in the illustration

Answer: Thank you very much for your review, we have changed in the manuscript.

  1. mTOR signaling pathway

Sometimes large letters are used unnecessarily in chemical names.

There is no reference to an illustration (Figure 3.).

Answer: Thank you very much for your review, we have changed in the manuscript.

  1. Figure 3.

C and D have to be swapped out.

NF-κB signaling pathway

Answer: Thank you very much for your suggestion, we have reorganized this chapter so that the figure have been removed from the manuscript.

  1. Nuclear factor-κB (NF-κB) transcription factors are master regulators of inflammation and immune homeostasis.

Transcription factors belonging to the family Nuclear Factor-κappa B (NF-κB) are master regulators of inflammation and immune homeostasis.

Answer:  Thank you very much for your suggestion, we have reorganized the chapter so this sentence has been removed.

  1. (Fig.3B) briefly describes the crosstalk relationship between Bmi-1 and NF-κB signaling

The crosstalk  between Bmi-1 and NF-κB signaling pathway briefly describes on Figure 3B.

Answer: Thank you very much for your suggestion, we have reorganized this chapter so that the figures have been removed from the manuscript.

  1. Inhibition of Bmi-1 inhibits osteosarcoma cell growth by inhibiting the expression of MMP9 and NF-κB signaling

Use synonyms

Answer: Thank you very much for your suggestion, we have reorganized the chapter so this sentence has been removed.

  1. Other signaling pathway

Other signaling pathways

Answer: Thank you very much for your suggestion, we have reorganized this chapter so this title has been removed.

  1. Figure 3D.

The interactions of Bmi-1, ink4a-ARF and c-MYC are not reflected in Figure 3D, while the other pathways shown in the picture are not described in the text.

Answer: Thank you very much for your suggestion, we have reorganized this chapter so that the figure have been removed from the manuscript.

  1. Notably, studies have shown that expression levels of Bmi-1 can contribute to the stabilization of YAP by blocking YAP in the nucleus

It is not clear from the text whether low or high levels of Bmi-1 expression stabilise YAP.

Answer: Thank you very much for your suggestion, we have reorganized the chapter so this sentence has been removed.

  1. Bmi-1 in cancer

This is an unfortunate title for a chapter, as the whole article is about Bmi-1 in cancer. You need to clarify the subject of the chapter

Answer: Thanks very much for your review, we have changed in the manuscript.

  1. The function of Bmi-1 in tumors has been identified in various pathological pro- cesses, including abnormal cell proliferation, evasion of apoptosis, tumor migration, and stemness maintenance of cancer stem cells (CSCs)

migration of cancer cells

Answer: Thanks very much for your review, we have changed in the manuscript.

Bmi-1 in cancer proliferation

  1. Mechanismly, when Bmi-1 is deficient, p16Ink4a is up-regulated, which prevents the binding of CDK4/6 to cyclinD, leading to phosphorylation of Rb[92]. Phosphorylated Rb binds to E2F to inhibit E2F-mediated tran- scription, which arrests the cell cycle in G0/G1 phase[102].

References 92 and 102 are mixed up.

Answer: Thanks very much for your review, we have changed in the manuscript.

  1. For example, Bmi-1-shRNA reduces the expression of cyclin D1 and increases the expression of p21/p27 through the INK4a-p16 independent pathway to arrest lung adenocarcinoma cells in G0/G1 phase[107].

There is no example in this sentence to illustrate the above

Answer: Thanks very much for your review, we have changed in the manuscript.

  1. Interestingly, Bmi-1 can interact with c-MYC to coordinately regulate progression of cell cycle and promote tumor formation[113]

If you are talking about protein-protein interactions between Bmi-1 and c-MYC and not about the regulation of transcription as discussed in the previous chapters, this needs to be dealt with separately

Answer: Thanks very much for your review, we have changed in the manuscript.

  1. Bmi-1 in cancer apoptosis

It is very difficult to draw conclusions from this chapter - which abnormalities in Bmi-1, i.e. an increase or decrease in expression, lead to a disruption of apoptosis.  If the different tumour types differ in this parameter - you need to list the variants so that practitioners can take advantage of this

Answer: Thanks very much for your review, we have supplemented in the manuscript.

Bmi-1 in cancer DNA damage response

  1. The lack of Bmi-1 in cells will cause mitochondrial dys- function, and at the same time, the DDR pathway will be initiated with the increase of ROS

Is the use of the future tense justified?

Answer: Thank you very much for your suggestion, we have corrected it in the manuscript.

  1. Mechanically, Bmi-1 binds to ring2/ring1b subunit to form a functional E3 ubiquitin ligase, and inhibits the expression of multiple gene through monoubiquitination of histone H2A in lysine 119 and Lys120 (Lys118 and Lys119 in H2A)[129]

Which lysines are modified?

Answer: Thank you very much for your review, we have corrected it in the manuscript.

Bmi-1 in cancer stem cells

  1. Stem cell has self-renewal ability, and produce at least one highly differentiated cell. Cancer stem cells (CSCs) refer to cancer cells with stem cell properties.

What properties are you talking about? You need to be more specific.

Answer: Thanks for your review, we have made a detailed description in the manuscript.

  1. 7 Others

Unfortunate title of the chapter. "Others" what?

Answer: Thanks very much for your review, we have changed in the manuscript.

  1. PTC- 209 increases the expression of DKK1 by down-regulating Bmi-1, impairs in vitro osteoclast formation and destroys the tumor microenvironment

Selective BMI-1 inhibitor PTC- 209 increases the expression of DKK1 by down-regulating Bmi-1, impairs in vitro osteo- clast formation and destroys the tumor microenvironment

Answer: Thank you very much for your review, we have corrected it in the manuscript.

  1. Clinical characteristics and cancer therapy of Bmi-1

6.1Protein expression and clinical characteristics of Bmi-1

Where is 6.2?

It is not a good idea to have the chapter as a table. It is better if the table and the small accompanying text form part of the next chapter, the title of which should be changed to reflect this part.

Answer: Thank you very much for your suggestions, we have merged them.

Bmi-1 in chemoresistance and cancer Therapy

  1. Interference of Bmi-1 in cancer cells enhances Camptothecin-induced DNA double-strand breaks and promotes Camptothecin-induced

What kind of interference are you talking about?

Answer: Thanks very much for your review, we have supplemented in the manuscript.

The chapter does not cover other Bmi-1 inhibitors:

PTC596 (Flamier, Anthony, et al. "Off-target effect of the BMI1 inhibitor PTC596 drives epithelial-mesenchymal transition in glioblastoma multiforme." NPJ precision oncology 4.1 (2020): 1-10.)

RU-A1 (Bartucci, Monica, et al. "Synthesis and characterization of novel BMI1 inhibitors targeting cellular self-renewal in hepatocellular carcinoma." Targeted oncology 12.4 (2017): 449-462.)

PTC-028 (Dey, Anindya, et al. "Evaluating the mechanism and therapeutic potential of PTC-028, a novel inhibitor of BMI-1 function in ovarian cancer." Molecular cancer therapeutics 17.1 (2018): 39-49.)

Answer: Thank you very much for your comments, we have added more sentences and literature to describe the inhibitors, and we have drawn a table for a more intuitive description.

Future research and conclusions

  1. In recent years, PcG family member Bmi-1 plays a vital role in proliferation, apopto sis, metastasis, chemical sensitivity of cancer cells.

Why only in recent years?

Answer: Thanks for your review, we have changed in the manuscript.

  1. While MiRNA mainly inhibits the expression of Bmi-1 by targeting the 3'UTR of Bmi-1[24]. For instance, the miRNA can inhibit the expression of Bmi-1 and then affect the proliferation, migration, invasion, apoptosis, and drug sensitivity of cancer

Many miRNAs inhibit Bmi-1 expression by targeting the 3'UTR [24-80]. That means, these miRNAs can affect cancer proliferation, migration, invasion, apoptosis and drug sensitivity [26, 55, 63].

Answer: Thanks for your review, we have changed in the manuscript.

  1. As a proto-oncogene, Bmi-1 has been confirmed to be highly expressed in a variety of tumors, and it is related to the clinical stage, pathological classification, lymph node metastasis and other malignancy of tumors, and can be used as one of the predictors of prognosis and recurrence of tumor patients.

The parameters listed fall into different logical categories.

Answer: Thank you very much for your review, we have corrected it.

  1. With the development of molecular biology technology and the application of gene chip technology, it is expected to develop a micro- fluidic multi-indicator joint inspection chip, a circulating cancer cell capture chip, and devices for automatic detection of Bmi-1 expression.

Need links to these new technologies

Answer: Thank you very much for your review, we have added references in the manuscript.

Reviewer 2 Report

Comments to the authors

The article with the title “The crucial roles of Bmi-1 in cancer:Implications in pathogenesis, metastasis, drug resistance, and targeted therapies” is in generally well done, but I would offer these comments to the investigators: 

1)      Several words throughout the manuscript appear to be merged. Please correct it.

2)      Some minor grammatical errors occur. The manuscript contains significant language-related issues. Please correct these types of grammatical errors throughout the paper.

3)      Introduction: “Therefore, Bmi-1 also is an oncogene, and its abnormal expression is associated with tumorigenesis and the drug resistance of many cancers [7].” Please provide the types of cancers that Bmi-1 expression has been associated with Tumorigenesis and drug resistance.

4)      Section 4.4. PI3K/Akt signaling pathway. Bmi-1 appears to regulate the PI3K/AKT/mTOR pathway. This signalling pathway inhibits autophagy, a homeostatic mechanism which regulates Tumorigenesis in several types of cancers. It will be interesting to mention the association of autophagy with Bmi-1 [PMID: 27050456, PMID: 25925206 PMID: 30487948, PMID: 33652741, PMID: 34721764 , PMID: 34947835]

5)      I strongly recommend a table with agents that modulate (inhibitors and/or activators) Bmi-1 operation.

Author Response

  • Several words throughout the manuscript appear to be merged. Please correct it.

Answer one: Thank you very much for your review, we have made corrections in the corresponding parts of the manuscript.

  • Some minor grammatical errors occur. The manuscript contains significant language-related issues. Please correct these types of grammatical errors throughout the paper.

Answer two:  Thanks for your review, we have changed in the manuscript.

  • Introduction: “Therefore, Bmi-1 also is an oncogene, and its abnormal expression is associated with tumorigenesis and the drug resistance of many cancers [7].” Please provide the types of cancers that Bmi-1 expression has been associated with Tumorigenesis and drug resistance.

Answer three: Thanks for your review, we have supplemented in the manuscript.

  • Section 4.4. PI3K/Akt signaling pathway. Bmi-1 appears to regulate the PI3K/AKT/mTOR pathway. This signalling pathway inhibits autophagy, a homeostatic mechanism which regulates Tumorigenesis in several types of cancers. It will be interesting to mention the association of autophagy with Bmi-1 [PMID: 27050456, PMID: 25925206 PMID: 30487948, PMID: 33652741, PMID: 34721764 , PMID: 34947835]

Answer four:  Thanks for your review, we have supplemented in the manuscript.

  • I strongly recommend a table with agents that modulate (inhibitors and/or activators) Bmi-1 operation.

Answer fiveThanks for your review, we have supplemented in the manuscript.

Round 2

Reviewer 1 Report

Dear Authors, 

The quality of your review article has increased and I think it can be published. Note table1: since all listed miRs down-regulate Bmi-1 the column "Effect on Bmi-1" is unnecessary, you can indicate the down-regulation in the title of the table. I recommend you to specify synonyms for Bmi-1 in the keywords and in the Introduction

Author Response

The quality of your review article has increased and I think it can be published. Note table1: since all listed miRs down-regulate Bmi-1 the column "Effect on Bmi-1" is unnecessary, you can indicate the down-regulation in the title of the table.

Answer: Thank you very much for your review, we have deleted the column and changed the title of the table.

 I recommend you to specify synonyms for Bmi-1 in the keywords and in the Introduction.

Answer: Thank you very much for your review, we have supplemented in the manuscript.

Reviewer 2 Report

The authors addressed to all of my concernes.

Author Response

Thank you very much for your suggestion and recognition.